# Practice of hyperglycaemia control in intensive care units of the Military Hospital, Sudan—Needs of a protocol

Ghada Omer Hamad Abd El-Raheem[1,2]*, Mudawi Mohammed Ahmed Abdallah[3], Mounkaila Noma[4]

1 Intensive Care Unit, Military Hospital, Khartoum, Sudan, 2 University of Medical Sciences and Technology UMST, High Diploma in Research Methodology and Biostatistics, Khartoum, Sudan, Khartoum, Sudan, 3 Intensive Care Unit, Military Hospital, Medical Manager of Critical Care Department, Military Hospital, Omdurman, Khartoum, Sudan, 4 University of Medical Sciences and Technology, Khartoum, Sudan

* ghadaomer90@gmail.com

**Data Availability Statement:** The minimal data set underlying the study has been provided in the paper and supporting information files. The hyperglycemia control local protocol is available at

## Abstract

Hyperglycaemia is a major risk factor in critically ill patients leading to adverse outcomes and mortality in diabetic and non-diabetic patients. The target blood glucose remained controversial; this study aimed to contribute in assessing the practice of hyperglycaemia control in intensive care units of the Military Hospital. Furthermore, the study proposed a protocol for hyperglycaemia control based on findings. A hospital-based cross-sectional study assessed the awareness and practice towards hyperglycaemia management in a sample 83 healthcare staff selected through stratified random sampling technique. In addition, 55 patients were enrolled, through quota sampling, after excluding those with diabetic ketoacidosis, hyperosmolar-hyperglycaemic state and patients < 18 years. A self-administrated questionnaire enabled to collect data from health staff and patient data were extracted from the medical records. SPSS-23 was used to analyze the collected data. Chi-square and ANOVA tests assessed the association among variables, these tests were considered statistically significant when $p \leq 0.05$. The training on hyperglycaemia control differed ($p = 0.017$) between doctors and nurses. The target glycaemic level (140–180 mg/dl) was known by 11.1% of the study participants. Neither the knowledge nor the practice of hyperglycaemia control methods differed among staff ($p > 0.05$). The use of sliding scale was prevalent (79.3%) across the ICUs ($p = 0.002$). 31.5% of the patients had received different glycaemic control methods, 11.8% were in the targeted blood glucose level. Sliding scale was the method used by doctors and nurses (71.4% and 81.6% respectively). Lack of awareness about hyperglycaemia management methods was prevalent among ICU healthcare staff. Use of obsolete methods was the common practice in the ICUS of the Military Hospital. Target blood glucose for patients were unmet. Development of a local protocol for glycaemic control in all ICUs is needed along with sustained training programs on hyperglycaemia control for ICU healthcare staff.

protocols.io https://www.protocols.io/file/
gre7brbrx.pdf?X-Amz-Algorithm=AWS4-HMAC-
SHA256&X-Amz-Credential=
AKIAIWSNCI5SNCPTWTQQ%2F20220303%2Fus-
east-1%2Fs3%2Faws4_request&X-Amz-Date=
20220303T171946Z&X-Amz-Expires=604800&X-
Amz-SignedHeaders=host&X-Amz-Signature=
966d71e97d5f44a4607d667a7b
595454faaff4dd76c5a08f0cb198205084d276.

**Funding:** The authors received no specific funding
for this work.

**Competing interests:** The authors declared no
competing interest.

**Abbreviations:** AACE, American Association of
Clinical Endocrinologists; ACP, American College of
Physicians; ADA, American Diabetes Association;
BG, Blood glucose; CCR, Critical Care Room; CCU,
Cardiac Care Unit; CGM, Continuous Glucose
Monitoring; ICU, Intensive Care Unit; IIT, Intensive
Insulin Therapy; NICE-SUGAR, The
Normoglycaemia in Intensive Care Evaluation—
Survival Using Glucose Algorithm Regulation;
NUH, Nottingham University Hospitals; POC, point
of care; SCCM, Society of Critical Care Medicine;
SGC, Space Glucose Control.

# Introduction

Hyperglycaemia is a major risk factor affecting critically ill patients leading to adverse out-
comes and a high mortality in diabetic and non-diabetic patients [1–4]. Stressful situations, as
acute illness and surgery in particular neurosurgery, elevate the levels of stress hormones and
increase hepatic glucose production, lipolysis and insulin resistance [5–7]. The stress cascade
increases by 7–8 folds in patients undergoing surgery [8,9] leading to more than three folds
increase in post-surgical complications and by six folds for mortality [7]. The target blood glu-
cose (BG) had been controversial. Leuven 1 study was the first landmark clinical trial that
revealed the benefits of reduced morbidity and mortality related to intensive insulin therapy
(IIT) in surgical critically ill patients [3,10]. However, the second Leuven trial with a higher
hypoglycaemia rate, pointed out that in medical intensive care unit (ICU) patients, there was a
no statistically significant difference in mortality rate between tight blood glucose group (80–
110 mg/dl) and control group [2,3,10]. In 2009, the practice changed following the publication
of the Normoglycaemia in Intensive Care Evaluation (NICE) and Survival Using Glucose
Algorithm Regulation (SUGAR) trial [11]. It revealed that the mortality and the hypoglycae-
mia increased in the intensive insulin therapy group compared to the conventional group. Fur-
thermore, a subgroup analysis indicated a no difference in outcomes between medical,
surgical, diabetic, non-diabetic and septic patients [2,3,7]. With respect to these results, con-
ventional target of < 180 mg/dl was acceptable for most ICU patients [4] and adopted by vari-
ous professional organizations [2,3,12,13], except the American College of Physicians (ACP)
which proposed a higher target of BG (< 200 mg/dl) [14]. These controversial benefits from
intensive insulin therapy should not shadow the reduction of complications and length of hos-
pitalization in hepatobiliary-pancreatic surgical patients [15]. Intravenous insulin through
infusion pump is the method applied for ICU patients [11,13]. The concomitant use of sub-
cutaneous insulin glargine remains more efficacious than insulin infusion alone, in particular
in patients with coronary artery bypass graft [16,17].

For policy development, it is crucial to assess the current practice by surveying both health-
care staff and patients to identify barriers and facilitators through a gap analysis to establish
the best practice [18]. Regarding hyperglycaemia control policy, the major safety issue
remained hypoglycaemia, especially in ICU patients, as the usual symptoms might not be
noticed [19]. Hypoglycaemia defines as blood glucose level < 70 mg/dl and severe life threat-
ening when it is < 40 mg/dl [11,20,21].

Protocols were developed as written instructions to prevent the fluctuation in BG due to
changes of interventions as administering steroids, vasopressors or parenteral quinine or due
to changes in nutrition support [22]. As hyperglycaemia is more prevalent in patients receiving
parenteral nutrition [9], BG levels did not differ among eating and non-per oral (NPO)
patients [23]. The protocols had differences in their target BG, monitoring frequencies, infu-
sion rates and use of boluses [24]. Hence, they must be customized to suit local resources, staff
competency [25] and the needs of patients [25–27]. Examples of these protocols are Portland l,
Washington University, and Yale University protocols. Yale Protocol had more difficult calcu-
lations than the other protocols [28], however, its hypoglycaemia rate was lower than Leuven
protocol [22]. The Nottingham University Hospitals (NUH) protocol adopted BG target levels,
which were consistent with the NICE-SUGAR target [13].

Alternative approaches to written policies are computerized protocols such as glucomman-
ders [28], star protocol [4] and space glucose control (SGC) system [29]. Although, they
reduced the nursing workload and had lower hypoglycaemia rates [21], they had not changed
the general practice [27].

This study aimed to assess the practice of healthcare staff towards hyperglycaemia control in intensive care units of the Military Hospital in oder to identify the gaps in knowledge and practice. This study proposed a protocol for hyperglycaemia control from the lessons learned.

## Materials and methods

A hospital- based cross-sectional study assessed the awareness, and practice of healthcare staff towards hyperglycaemia management and the burden of hyperglycaemia control based medical records of critically ill patients in the intensive care units of the Military Hospital of Khartoum State, Sudan. The Military Hospital is a complex of seven specialized hospitals totalizing 722 beds and 8 ICUs. A multistage sampling technique was used. At first level, five ICUs were systematically included in the study after excluding the neonatal, the maternity and the medical ICUs the last being under reconstruction. At second level, a stratified random sampling technique enabled to select 83 health professionals (doctors and nurses) proportionally to the size of each ICU after excluding the administrative staff. Regarding the patients included in the study, a quota of 12 patients was fixed to randomly recruit participants from each of the five ICUs. This led to an estimated sample of 60 patients. Fifty-five patients were enrolled in the study after excluding those with either diabetic ketoacidosis (DKA) [5] or hyperosmolar hyperglycaemic state (HHS) and patients < 18 years. Patients with DKA or HHS were excluded because the random blood glucose of such patients is extremely high and fluctuating which might lead to inaccurate presentation of the hyperglycemia status in the ICU. However, because of the importance of such disorders, DKA was included in the assessment of the knowledge and practice of healthcare staff. Data were collected through a standardized questionnaire comprising two parts. Part one was a self-administered questionnaire filled by the healthcare staff working in ICUs to collect their sociodemographic characteristics, their number of years of working experience, their knowledge and practice on hyperglycaemia control methods and levels as well as the management of hyperglycaemia. Part two extracted data from the medical records of ICU patients hospitalized at the time of the data collection. The characteristics of the patients: age, gender, status (medical or surgical), type of hyperglycaemia (diabetes type 1, 2 or non-diabetic), associated comorbidities, methods of blood glucose measurement and levels were recorded. SUMASRI Institutional Review Board of the University of Medical Sciences and Technology reviewed the proposal in an expedited review board and gave the ethical clearance to conduct this study with no ethical restrictions because the study had no any harm to any of the participants because there were no any medical tests or procedure were done specifically for the study, and that the study relied only on the coded responses of doctors and nurses and the coded medical records of the patients without any identity exposure. Ethical Approval was obtained from the Military Hospital, the implementation of the research was granted by the administration of the respective ICUs. Participants (doctors, nurses) were well informed about the research objectives and verbal informed consent was obtained from each one of them and then each participant filled the self-administered questionnaire. Only the participants who approved to participate filled the questionnaire. As for the patients, informed consents were obtained from the surrogate decision makers of the critically ill patients prior to extracting the medical information from the patients' files. They were ensured about their confidentiality with the use of an anonymous research tool and that the data collected from them would be used strictly for the purpose of the study objectives.

The statistical package for social sciences (SPSS version 23) was used to describe and analyse the data. Statistical analysis performed were chi-square tests and analysis of variance (ANOVA) to determine association among variables. All tests were considered statistically significant when $p < 0.05$.

## Results

### Characteristics of healthcare staff and their training on hyperglycaemia control

The majority (74.1%, 60/81) of the participants were nurses and the remaining 25.9% (21/81) were doctors. 77.8% (63/81) of the participants were aged 25–30 years with no statistical association ($p = 0.05$) between the age of the participants and their occupation. The years of working experience of the participants ranged between 0.1 year and 12 years with a median of 1 year; while, working years in intensive care unit ranged from 0.01 years to 8 years with a median of working years of 0.5 years. 66.7% (14/21) of the doctors received training on hyperglycaemia control and 36.7% (22/60) of the nurses were trained with a statistically significant difference ($\chi^2 = 5.67$, $p = 0.017$) between the status and being trained on hyperglycaemia, Table 1.

### Awareness of healthcare staff about the target blood glucose level

The 81 healthcare staff were asked if they knew the target blood glucose (BG) level, 88.9% (72/81) replied yes, 27.8% (20/72) of them were doctors and 72.2% (52/72) were nurses. They were 9 who did not know (1 doctor and 8 nurses). There was a no statistically significant association (Likelihood ratio = 1.349, $p = 0.245$) between the awareness about target BG level and the staff status. However, when prompted to provide the exact level of the target blood glucose, they were 11.1% (8/72) who provided the correct level (140–180 mg/dl) and 88.9% (64/72) reported incorrect levels. Of the eight participants who reported the correct level, 62.5% (5/8) were doctors and 37.5% (3/8) were nurses. A statistically significant difference (Fisher's Exact Test, $p = 0.033$) was found between the reported level of blood glucose and the status of the healthcare staff.

**Table 1. Characteristics of the healthcare staff and training on glycaemic control (n = 81).**

| Characteristics | Status of the staff | | | | | | Likelihood ratio | *p*-value |
|---|---|---|---|---|---|---|---|---|
| | Doctor | % | Nurse | % | Total | % | | |
| **Age:** | | | | | | | | |
| 25–30 years | 13 | 20.6 | 50 | 79.4 | 63 | 77.8 | 3.835 | 0.05 |
| > 30 years | 8 | 44.4 | 10 | 55.6 | 18 | 22.2 | | |
| **Total (%)** | **21** | **25.9** | **60** | **74.1** | **81** | **100.0** | | |
| **Gender:** | | | | | | | | |
| Female | 19 | 28.8 | 47 | 71.2 | 66 | 81.5 | 1.697 | 0.193 |
| Male | 2 | 13.3 | 13 | 86.7 | 15 | 18.5 | | |
| **Total (%)** | **21** | **25.9** | **60** | **74.1** | **81** | **100.0** | | |
| **ICU working experience:** | | | | | | | | |
| <1year | 11 | 20.4 | 43 | 79.6 | 54 | 66.7 | 4.849 | 0.089 |
| 1–3 years | 9 | 45.0 | 11 | 55.0 | 20 | 24.7 | | |
| >3 years | 1 | 14.3 | 6 | 85.7 | 7 | 8.6 | | |
| **Total (%)** | **21** | **25.9** | **60** | **74.1** | **81** | **100.0** | | |
| **Training about Glycaemic control:** | | | | | | | | |
| Trained | 14 | 38.9 | 22 | 61.1 | 36 | 44.4 | 5.67* | 0.017 |
| Untrained | 7 | 15.6 | 38 | 84.4 | 45 | 55.6 | | |
| **Total (%)** | **21** | | **60** | | **81** | **100.0** | | |

*chi-square test.

### Awareness of healthcare staff about Basal-Bolus and insulin infusion methods

Regarding the awareness of healthcare staff about the hyperglycaemia control methods, 27 (6 doctors and 21 nurses) out of the 81 participants were unaware of the basal bolus method. While, 67 (16 doctors and 51 nurses) were unaware of the insulin infusion method. There was no statistically significant association between the knowledge of the staff about Basal-Bolus and Insulin Infusion, and their training status ($p = 0.591$ and $0.371$ respectively). Lack of knowledge was the main reason reported by 96.3% and 97% of the staff regarding these two hyperglycaemia control methods.

### Practice of healthcare staff towards blood glucose monitoring

The practice of staff towards blood glucose (BG) measurement was assessed as either more frequently ($< 6$ hourly) or less frequently ($\geq 6$ hourly). 47.6% of the doctors measured BG more frequently and 52.4% measured BG level less frequently. With regards to nurses, 35.0% measured BG more frequently and 65.0% measured it less frequently. In the overall, a no statistically significant association was found ($\chi^2 = 2.197$, $p = 0.138$) between the training of staff and their practice towards BG monitoring frequency. Across the three types of intensive care units, *HbA1c* was requested by 69.5% of the staff. In cardiac care unit (CCU), the request was from all (8/8) the staff; while in mixed and surgical ICUs it was respectively from 71.9% and 30% of the staff. There was a statistically significant association (Likelihood ratio = 12.584, $p = 0.002$) between ICU type and the request for *HbA1c*.

### Practice of healthcare staff towards diabetic ketoacidosis

In the overall, the appropriate management of diabetic ketoacidosis (DKA), consisting of overlapping the I.V and S.C insulin was performed by 29.6% of the participants. The remaining 70.4% either they stop the I.V insulin before starting the S.C insulin (56.8%)) or they did not know what to do (13.6%); with a statistically significant difference between doctors and nurses (Likelihood ratio = 10.2, $p = 0.006$).

### Hyperglycaemia control methods used by healthcare staff in the different intensive care units

*Sliding scale method* was used by 90% (9/10) of the surgical ICU staff, 84.4% (54/64) of the mixed and 25% (2/8) of the cardiac ICU. In the overall, across these three types of ICU, sliding scale method was used by 79.3% of the staff and they were 20.7% (17/82) who used other methods. These other methods used were Basal-Bolus method (82.3%), mixed insulin method (11.8%) and insulin infusion method (5.9%). There was a statistically significant association (Likelihood ratio = 12.728, $p = 0.002$) between the use of sliding scale method and the type of the ICU. Fig 1 revealed the distribution of staff by hyperglycaemia method used.

Usual practice was the main reason (80.5%) reported by the participants for using sliding scale method, of the staff reported that it was the usual practice (p = 0.000) and more than half (53.7%) were not satisfied with the control method they were using.

### Number of infusion pumps per ICU

The number of infusion pumps available per ICU patient ranged from 0 to 6 with a statistically significant mean number infusion pumps of $2.95 \pm 1.33$ ($p = 0.001$) across the five ICUs (3 mixed, 1 cardiac and 1 surgical). The lowest mean number of infusion pumps was recorded in the surgical unit (1.70 pumps±0.82, [range: 0–3]).

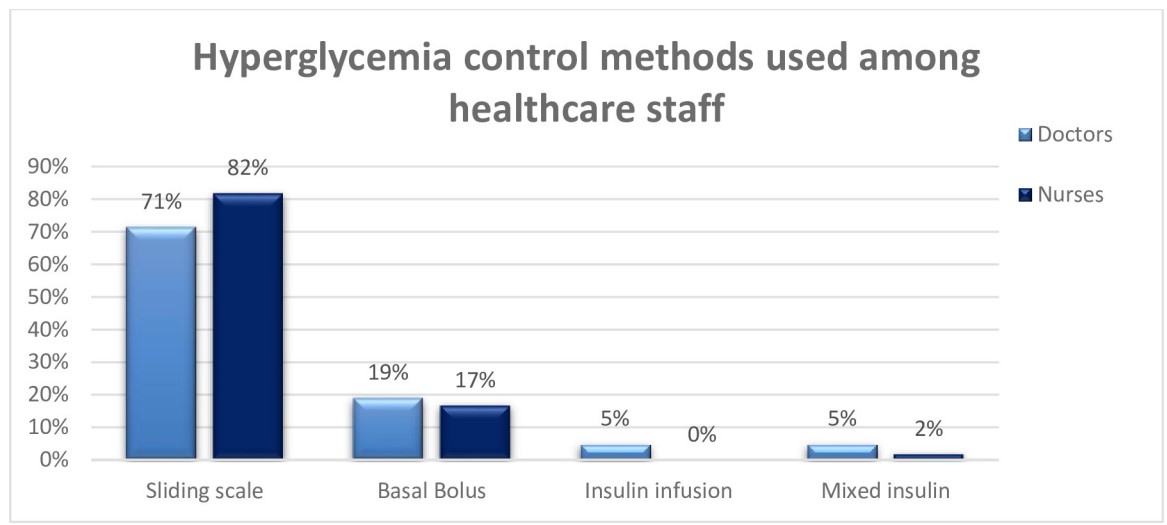

**Fig 1. Hyperglycaemia control methods used among health care professionals.**

### Characteristics of ICU patients

Of the Fifty-five patients selected across the ICUs of the Military Hospital, 50.9% were males and 49.1% were females. They aged between 19 and 95 years with a median age of 63.5 years. More than half (58.2%, 32/55) were under enteral feeding. 72.8% (40/55) were non-diabetic; 23.6% (13/55) and 3.6% (2/55) were respectively type 2 and type 1 diabetic patients. Table 2 displayed the other characteristics of the patients.

### Glycaemic status of ICU patients and hyperglycaemia control methods used in Military Hospital

A statistically significant association (Likelihood ratio = 49.964, $p$ = 0.000) was found between the hyperglycaemia control method used and the diabetes status of the patients as indicated by Table 3.

### Blood glucose levels and hyperglycaemia control methods used

Two classifications of random blood glucose (*the glycaemic levels* and *the NICE-SUGAR* blood glucose levels) were used. The *glycaemic levels classification* revealed that 79.6% (43/54) of the patients were normal glycaemic (BG: 71–180 mg/dl), 18.5% (10/54) were hyperglycaemic (BG: > 180 mg/dl) and a patient (1.9%, 1/54) was hypoglycaemic (BG: < 71 mg/dl). In the other hand, the *NICE-SUGAR blood glucose classification* indicated that 61.1% (33/54) of the patients were below range (BG: <140 mg/dl), 20.4% (11/54) were in within random glucose level (BG: 140–180 mg/dl), 18.5% (10/54) were above the range of BG > 180 mg/dl.

Regarding the hyperglycaemia control methods, they were used for 31.5% (17/54) of the patients. Table 4 revealed that 5.9% (1/17) of the patients was monitored in using the best appropriate method which was insulin infusion; 29.4% (5/17) of the patients were under alternative glycaemia control methods which were namely basal- bolus (11.8%, 2/17), mixed insulin (11.8%, 2/17) and Oral (5.9%, 1/17). Unfortunately, the majority of patients (64.7%, 11/17) had their glycaemia control based on the old fashion method of sliding scale despite a no statistically significant association (likelihood ratio = 10.108, $p$ = 0.258) between the NICE- SUGAR targets and the method used, Table 4.

**Table 2. Characteristics of the patients hospitalized in the ICUs of the Military Hospital (n = 55).**

| Variable | n | % | Variable | n | % |
|---|---|---|---|---|---|
| **Gender (n = 55)** | | | **Renal function (n = 55)** | | |
| Male | 28 | 50.9 | Normal | 33 | 60.0 |
| Female | 27 | 49.1 | Impaired | 22 | 40.0 |
| **Age in years (n = 54)** | | | **Liver function (n = 55)** | | |
| Median | 63.5 | | Normal | 53 | 96.4 |
| Min-Max | 19–95 | | Impaired | 2 | 3.6 |
| **Patients conditions (n = 55)** | | | **Patients on vasopressors (n = 55)** | | |
| Sepsis | 17 | 30.9 | No | 46 | 83.6 |
| Neurological | 8 | 14.5 | Yes | 9 | 16.4 |
| Cardiovascular | 8 | 14.5 | **Patients on steroids (n = 55)** | | |
| Trauma | 8 | 14.5 | No | 43 | 78.2 |
| Stroke | 5 | 9.1 | Yes | 12 | 21.8 |
| Gastroenterology | 3 | 5.5 | **Patients on quinine I.V. (n = 55)** | | |
| Cancer/Tumor | 2 | 3.6 | No | 54 | 98.2 |
| Endocrine | 2 | 3.6 | Yes | 1 | 1.8 |
| Respiratory | 2 | 3.6 | **On Fluoroquinolones (n = 55)** | | |
| **Hyper glycaemia status (n = 55)** | | | No | 51 | 92.7 |
| Non diabetic | 40 | 72.8 | Yes | 4 | 7.3 |
| Type 2 DM | 13 | 23.6 | **On atypical antipsychotics (n = 55)** | | |
| Type 1 DM | 2 | 3.6 | No | 54 | 98.2 |
| **Feeding status (n = 55)** | | | Yes | 1 | 1.8 |
| Enteral feeding | 32 | 58.2 | | | |
| Oral feeding | 15 | 27.3 | | | |
| Non per oral | 8 | 14.5 | | | |

## Discussion

Training of health professionals is crucial to sustain evidence-based practice [30,31]. More than half (66.7%, 14/21) of the doctors received training on hyperglycaemia control, while, only 36.7% (22/60) of the nurses did ($p = 0.017$). Interestingly, there was no difference ($p > 0.05$) in knowledge between doctors and nurses about Basal-Bolus and insulin infusion methods and their training status. This emphasized the need for a standard updated policy with appropriate training material addressing the gaps of knowledge on hyperglycaemia control methods regardless the status of the staff [18]. Moreover, the practice of our staff towards blood glucose monitoring frequency using ICU lab and point of care method did not differ between trained and untrained

**Table 3. Hyperglycaemia control methods by the glycaemic status of the patients (n = 55).**

| Method used | Hyperglycaemia status | | | | | | Total | | p-value* |
|---|---|---|---|---|---|---|---|---|---|
| | Type 1 DM | | Type 2 DM | | Non-diabetic | | | | |
| | n | % | n | % | n | % | n | % | |
| Sliding scale | 0 | 0.0 | 9 | 81.8 | 2 | 18.2 | 11 | 20.0 | 0.000 |
| Basal-Bolus | 0 | 0.0 | 2 | 100.0 | 0 | 0.0 | 2 | 3.6 | |
| Insulin I.V infusion | 0 | 0.0 | 1 | 100.0 | 0 | 0.0 | 1 | 1.8 | |
| Other methods | 1 | 33.3 | 1 | 33.3 | 1 | 33.3 | 3 | 5.5 | |
| None | 1 | 2.6 | 0 | 0.0 | 37 | 97.4 | 38 | 69.1 | |
| **Total patients** | **2** | **3.6** | **13** | **23.6** | **40** | **72.7** | **55** | **100.0** | |

* Likelihood ratio = 49.964.

**Table 4. Glycaemic levels of patients by hyperglycaemia control methods (n = 17).**

| Hyperglycemia control method | NICE-SUGAR blood glucose levels | | | | | | | | *p*-value* |
|---|---|---|---|---|---|---|---|---|---|
| | Above range (BG>180mg/dl) | | In range (BG 140-180mg/dl) | | Below range (BG<140mg/dl | | Total | | |
| | n | % | n | % | n | % | n | % | |
| Sliding scale | 4 | 36.4 | 1 | 9.1 | 6 | 54.5 | 11 | 64.7 | 0.258 |
| Basal- Bolus | 1 | 50.0 | 0 | 0.0 | 1 | 50.0 | 2 | 11.8 | |
| Mixed insulin | 2 | 100.0 | 0 | 0.0 | 0 | 0.0 | 2 | 11.8 | |
| Oral (Glimepiride) | 1 | 100.0 | 0 | 0.0 | 0 | 0.0 | 1 | 5.9 | |
| Insulin infusion | 0 | 0.0 | 1 | 100.0 | 0 | 0.0 | 1 | 5.9 | |
| **Total patients** | **8** | 47.1 | **2** | 11.8 | **7** | 41.2 | **17** | 100.0 | |

* Likelihood ratio = 10.108.

doctors and nurses. This monitoring method using point of care (POC), glucometer or ICU laboratory, is acceptable as well as continuous glucose monitoring (CGM) [32].

Regarding the practice towards DKA, as published in the literature [1,19,33], it statistically differed between doctors and nurses ($p = 0.006$), as well as, according to training status ($p = 0.036$).

Consistent with guideline-based practice [34,35], all the staff (100.0%) of the cardiac care unit (CCU) in the Military Hospital were practicing the HbA1c measurement, as expected in such unit; contrary to the mixed (71.9%) and surgical ICUs (30.0%), ($p = 0.002$).

Assessment on barriers and facilitators on policy implementation is required for developing protocols [18,31], such as availability of infusion pumps (indicated for the administration of insulin) for ICU patients [13,34,35]. This was considered a barrier in our study with the surgical ICU being the least equipped (1.7 ±0.82infusion pumps).

The dominant hyperglycaemia control method in both surgical and mixed ICUs was sliding scale, which stood as the standard practice of our study participants while this method was discouraged [17,19,34,36,37]. Insulin infusion method is the recommended control method [1,28,22,37], hence the need to move away nowadays from sliding scale [36]. This was our leitmotiv for proposing a protocol for glycaemia control in Sudan ICUs, Military Hospital (supporting information). The proposed protocol is justified by our findings, which revealed that more than half of the care providers used sliding scale and were satisfied with it. The target blood glucose level of 140–180 mg/dl, acceptable for most ICU patients [11] and adopted by most of the major agencies [2,3,13] was known by only 11% of our study participants. This appeal for the adoption of local institutional guidelines for all Military Hospital ICUs given the diversity of the specialities of health professionals [34].

Regarding patients, sliding scale method was used for 20% of the patients, this was consistent with a Brazilian study reporting the dominant use of sliding scale in ICUs [38]. Blood glucose readings pointed that 11.8% of the patients had readings in the target range of 140–180 mg/dl, 41.2% had BG levels below the target range and 47.1% of the patients were hyperglycaemic (BG > 180 mg/dl). Our findings raised concerns about the nutritional status of the patients and the methods used as discussed in the literature [8,9,39].

In our study, insulin infusion method was used for one patient and the NICE-SUGAR target was achieved [6,11]. While, mixed insulin method did not achieve the target glycaemic range as already reported by Marik P.E et al. [10]. Sliding scale method achieved the target range in only 9.1% of the patients of our study; consistent with published literature [8,19,40] recommending the use of insulin infusions in ICU patients to achieve the NICE-SUGAR range which had proven efficacy and safety in low-income countries [41].

This study was not without limitations, the data collected from the working staff were not validated through Cronbach test of reliability. Nevertheless, the study was conducted in a single center with multiple ICUS. The study was based on the self-reporting of the doctors and nurses which might be inaccurate. Besides, the collection of the medical data of the patients relied on the quality of documentation.

## Conclusions

Lack of awareness about hyperglycaemia management methods was prevalent among ICU healthcare staff. Use of obsolete methods was the common practice in the ICUS of the Military Hospital. Target blood glucose for patients were unmet. Development of a local protocol for glycaemic control in all ICUs is needed along with sustained training programs on hyperglycaemia control for ICU healthcare staff.

## Supporting information

**S1 Table. Awareness of health care staff towards hyperglycaemia control methods and the reasons for lack of awareness (n = 81).**
(DOCX)

**S2 Table. Management of diabetic ketoacidosis (DKA) by the participants according to their status of training on glycaemia control (n = 81).**
(DOCX)

**S3 Table. Practice of measurement of blood glucose by health professional according to their training status (n = 81).**
(DOCX)

**S4 Table. Glycaemia measurement methods used the staff of the intensive care units.**
(DOCX)

**S5 Table. Glycaemia control methods used by healthcare professionals (n = 81).**
(DOCX)

**S6 Table. Reasons reported by ICU staff and their satisfaction for using a given glycaemia control method.**
(DOCX)

**S7 Table. Number of infusion pumps available for each patient across different ICUs.**
(DOCX)

**S8 Table. Glycaemic ranges and the hyperglycaemia methods used.**
(DOCX)

**S1 Dataset. Minimal data set.**
(XLSX)

**S1 File.**
(DOCX)

## Acknowledgments

The authors are grateful to the patients, doctors and nurses of the Military Hospital whose participation enabled this study to be completed.

## Author Contributions

**Data curation:** Ghada Omer Hamad Abd El-Raheem.

**Formal analysis:** Ghada Omer Hamad Abd El-Raheem.

**Investigation:** Ghada Omer Hamad Abd El-Raheem.

**Methodology:** Ghada Omer Hamad Abd El-Raheem.

**Supervision:** Mounkaila Noma.

**Validation:** Ghada Omer Hamad Abd El-Raheem.

**Writing – original draft:** Ghada Omer Hamad Abd El-Raheem.

**Writing – review & editing:** Ghada Omer Hamad Abd El-Raheem, Mudawi Mohammed Ahmed Abdallah, Mounkaila Noma.

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
