## [Decision Letter · Decision Letter 0]

3 Jan 2022

PONE-D-20-22767

Practice of Hyperglycemia Control in Intensive Care Units of the Military Hospital, Sudan- Needs of a Protocol

PLOS ONE

Dear Dr. Abd El-Raheem,

Thank you for submitting your manuscript to PLOS ONE. After careful consideration, we feel that it has merit but does not fully meet PLOS ONE’s publication criteria as it currently stands. Therefore, we invite you to submit a revised version of the manuscript that addresses the points raised during the review process.

We look forward to receiving your revised manuscript.

Kind regards,

Chiara Lazzeri

Academic Editor

PLOS ONE

Journal Requirements:

2. Please amend your current ethics statement to address the following concerns: Please explain why written consent was not obtained, how you recorded/documented participant consent, and if the ethics committees/IRBs approved this consent procedure.

3. Thank you for stating the following financial disclosure: "NA"

Reviewers' comments:

Reviewer's Responses to Questions

**Comments to the Author**

1. Is the manuscript technically sound, and do the data support the conclusions?

Reviewer #1: Yes

Reviewer #2: Partly

2. Has the statistical analysis been performed appropriately and rigorously? 

Reviewer #1: Yes

Reviewer #2: Yes

3. Have the authors made all data underlying the findings in their manuscript fully available?

Reviewer #1: Yes

Reviewer #2: Yes

4. Is the manuscript presented in an intelligible fashion and written in standard English?

Reviewer #1: Yes

Reviewer #2: Yes

5. Review Comments to the Author

Reviewer #1: This article is interesting and deals with a very topical issue, which has not yet been resolved. Despite the low numbers, the authors propose an alternative method to be evaluated on larger case series. The study was conducted with a valid methodology, not easily applicable to the context of intensive wards.

Reviewer #2: I read with interest the manuscript titled “Practice of Hyperglycemia Control in Intensive Care Units of the Military Hospital, Sudan- Needs of a Protocol”.

This study assessed the practice of healthcare staff on hyperglycaemia control in intensive care units of Khartoum Military Hospital.

While I think the study has many merits, some cissues need attention.

Results

The results are confusing and do not meet the study's objective; this should be simpler to maintain interest from the readers.

In the methods, the authors indicate that ketoacidosis patients were excluded, and, in the results, there is some analysis included.

Discussion

After reading the discussion, I still do not understand the study's objective.

The limitation should be better addressed and avoid any justification based on a change of awareness accomplished by the study's results, however meaningful; there are no part of the study's objectives.

Tables and figures

There are too many tables and figures; it would be essential to select the most representative and the others, if needed, add them as supplementary material.

Conclusion

The conclusion should be based on your results, the sentence in line 322, was not part of your analysis in the study, and it only represents a point of view from the authors.

6. PLOS authors have the option to publish the peer review history of their article (what does this mean?). If published, this will include your full peer review and any attached files.

Reviewer #1: No

Reviewer #2: **Yes: **Angel Augusto Perez-Calatayud

---

## [Author Response · Author response to Decision Letter 0]

13 Mar 2022

Dear Reviewer,

Thank you for dedicating you valuable time to conduct a critical appraisal for our submitted manuscript. It is our pleasure to submit to PLOS ONE the manuscript titled “Practice of Hyperglycaemia Control in Intensive Care Units of the Military Hospital, Sudan - Needs of a Protocol”. The authors received no funding for this work. We chose PLOS ONE to publish our manuscript, because of the powerful peer review process. We have addressed all the following points:

• Ethical concerns.

• Data availability.

• Organising the results to be more clear.

• Explaining why DKA blood glucose readings were excluded. 

• Rewriting the discussion and conclusion sections as well as the study limitations.

• Uploaded the figure to PACE tool and the hyperglycemia protocol to protocols.io 

All the supporting data were uploaded as supporting information with no legal or ethical restrictions.

Thank you again for your valuable time and priceless advices.

With our best regards, we remain.

Ghada Omer Hamad Abd El-Raheem

---

## [Editor Report · Decision Letter 1]

13 Apr 2022

Practice of Hyperglycemia Control in Intensive Care Units of the Military Hospital, Sudan- Needs of a Protocol

PONE-D-20-22767R1

Dear Dr. Abd El-Raheem,

We’re pleased to inform you that your manuscript has been judged scientifically suitable for publication and will be formally accepted for publication once it meets all outstanding technical requirements.

Kind regards,

Chiara Lazzeri

Academic Editor

PLOS ONE
---

## [Editor Report · Acceptance letter]

27 Apr 2022

PONE-D-20-22767R1 

Practice of Hyperglycaemia Control in Intensive Care Units of the Military Hospital, Sudan - Needs of a Protocol   

Dear Dr. Abd El-Raheem:

I'm pleased to inform you that your manuscript has been deemed suitable for publication in PLOS ONE. Congratulations! Your manuscript is now with our production department. 

Kind regards, 

on behalf of

Dr. Chiara Lazzeri 

Academic Editor

PLOS ONE